# Transcriptomic Signature of Horseshoe Crab *Carcinoscorpius rotundicauda* Hemocytes' Response to Lipopolysaccharides

**Maria E. Sarmiento [1], Kai Ling Chin [2], Nyok-Sean Lau [3], Noraznawati Ismail [4], Mohd Nor Norazmi [1], Armando Acosta [1,\*] and Nik Soriani Yaacob [5,\*]**

1    School of Health Sciences, Universiti Sains Malaysia, Health Campus, Kubang Kerian 16150, Malaysia
2    Faculty of Medicine and Health Sciences, Universiti Malaysia Sabah, Kota Kinabalu 88400, Malaysia
3    Centre for Chemical Biology, Universiti Sains Malaysia, Bayan Lepas 11900, Malaysia
4    Institute of Marine Biotechnology, Universiti Malaysia Terengganu, Kuala Nerus 21030, Malaysia
5    Department of Chemical Pathology, School of Medical Sciences, Universiti Sains Malaysia, Health Campus, Kubang Kerian 16150, Malaysia
\*    Correspondence: armando@usm.my (A.A.); niksoriani@usm.my (N.S.Y.)

**Abstract:** *Carcinoscorpius rotundicauda* (*C. rotundicauda*) is one of the four species of horseshoe crabs (HSCs). The HSC hemocytes store defense molecules that are released upon encountering invading pathogens. The HSCs rely on this innate immunity to continue its existence as a living fossil for more than 480 million years. To gain insight into the innate mechanisms involved, transcriptomic analysis was performed on isolated *C. rotundicauda* hemocytes challenged with lipopolysaccharides (LPS), the main components of the outer cell membrane of gram-negative bacteria. RNA-sequencing with Illumina HiSeq platform resulted in 232,628,086 and 245,448,176 raw reads corresponding to 190,326,253 and 201,180,020 high-quality mappable reads from control and LPS-stimulated hemocytes, respectively. Following LPS-stimulation, 79 genes were significantly upregulated and 265 genes were downregulated. The differentially expressed genes (DEGs) were related to multiple immune functional categories and pathways such as those of the cytoskeleton, Toll and Imd, apoptosis, MAP kinase (MAPK), inositol phosphate metabolism, phagosome, leucocyte endothelial migration, and gram-negative bacterial infection, among others. This study provides important information about the mechanisms of response to LPS, which is relevant for the understanding the HSCs' immune response.

**Keywords:** Carcinoscorpius rotundicauda; RNA-sequencing; lipopolysaccharides challenge; immune response

## 1. Introduction

Horseshoe crabs (HSCs) are incredible living fossils, existing for at least 480 million years, making them even older than the dinosaurs. There are four species of HSCs: *Tachypleus gigas* (*T. gigas*), *Tachypleus tridentatus* (*T. tridentatus*), and *Carcinoscorpius rotundicauda* (*C. rotundicauda*), which are found around the coasts in Asia, and *Limulus polyphemus* (*L. polyphemus*), which inhabits the eastern coast of North America and the Gulf of Mexico [1]. HSCs belong to the arachnid family tree, more related to spiders, scorpions, mites, and ticks than to crustaceans [2]. HSC hemolymph is unique; it is a copper-based blue blood that is the source of limulus amebocyte lysate (LAL), which is extremely important to the biomedical industry for testing vaccines, drugs, and medical devices for contamination with bacterial endotoxins [3]. The demand for the HSCs' blood and ecological destruction have contributed to the population decline. The International Union for Conservation of Nature (IUCN) Red List of Threatened Species has listed *T. tridentatus* as "Endangered", followed by *L. polyphemus* as "Vulnerable", and *C. rotundicauda* and *T. gigas* as "Data deficient" (http://www.iucnredlist.org, accessed on 17 November 2022).

The hemocytes are the main regulators of innate immunity in HSCs. When hemocytes are in contact with lipopolysaccharides (LPS), which are part of the outer cell wall of gram-

negative bacteria, defense molecules stored in the hemocytes are released, triggering the coagulation cascade and neutralization of the pathogens [4]. A study of *C. rotundicauda* after infection with *Pseudomonas aeruginosa* (*P. aeruginosa*) showed that the hemocytes respond to acute infections by gram-negative bacteria, activating immune genes including the synthesis, storage, and secretion of immune proteins and effectors to sustain the frontline innate immune defense [5].

High-throughput RNA-sequencing (RNA-seq) is a powerful method for profiling the transcriptome of a cell. The identified differentially expressed and co-regulated genes are important to inform probable biological function to understand the complex and dynamic nature of different physiological or pathological conditions [6]. We recently reported that multiple immune-related genes were differentially expressed in LPS-stimulated *T. gigas* hemocytes [7]. In the current study, we stimulated isolated HSC hemocytes of *C. rotundicauda* challenged with LPS to obtain information about the mechanisms of innate immune defense in this HSC species.

## 2. Materials and Methods

### 2.1. Animals

Three adult *C. rotundicauda* HSCs were obtained from Kuala Kemaman, Terengganu, located at the east coast of Peninsular Malaysia. The HSCs were transported at night to a hatchery at the Institute of Marine Biotechnology, Universiti Malaysia Terengganu, and were allowed to adapt to the new conditions for three days prior to hemocyte isolation.

### 2.2. LPS Challenge

Hemolymph (2 mL) was collected from each animal, by trained technicians under sterile conditions, using pyrogen-free materials in a Biological Safety Cabinet Class II (ESCO, USA). The HSCs were then returned to the sea.

The LPS challenge was performed as previously described [7], based on the report of Ozaki et al., 2005 [8] with several modifications. The hemolymph (2 mL) from each HSC was mixed with 3% NaCl (25 mL), and each well of a six-well cell culture plate was filled with the suspension (2 mL) and incubated for 15 min at room temperature (25 °C). The supernatant of each well was discarded and $10^{-13}$ g/mL of *E. coli* LPS (Sigma, USA) in 3% NaCl (2 mL) was added to the wells for stimulated cultures (CrLPS), while 3% NaCl (2 mL) was added for the non-stimulated cultures (CrNS), both in triplicates. After 1 h incubation at room temperature (25 °C), the supernatant was discarded before proceeding to RNA extraction.

### 2.3. RNA Extraction and Sequencing

Triplicate RNA samples from the attached hemocytes of each HSC (CrLPS and CrNS) were extracted using the Nucleospin RNA Mini Kit (Macharey-Nagel, Duren, Germany) according to the manufacturer's protocols. The three RNA samples from CrLPS of each animal were pooled and the same procedure was carried out with the CrNS samples. The quantity and quality of total RNA were assessed using Qubit 2.0 Fluorometer (Life Technologies, Carlsbad, CA, USA) and Agilent 2100 Bioanalyzer (Agilent Technologies, Santa Clara, CA, USA). Six sequencing libraries from CrLPS and CrNS of the three HSCs were obtained. Sequencing libraries were prepared from 0.4 μg total RNA using a Truseq^TM RNA sample prep kit (Illumina, San Diego, CA, USA) and were sequenced on an Illumina HiSeq 4000 system [7].

### 2.4. RNA-Seq Analysis

The raw reads were trimmed with TrimGalore (v0.6.5) to filter low-quality bases and adaptor sequences. Then, the reads were mapped to *C. rotundicauda* reference genome (VWRL01) [9] using HISAT2 (v2.1.0) with option "—dta". The unique reads mapped to the exon were counted with python package HTSeq (v0.12.4) [10]. The edgeR R package (v3.24.1) was used for differential expression analysis [11]. The data were adjusted for

the batch effect, normalized using the trimmed mean of M-values (TMM) method and transformed to log2 counts per million (CPM) in edgeR. DEGs were identified using an adjusted *p*-value corrected by a false discovery rate (FDR) cut-off of 0.05, and log-fold change of >0.3 or <−0.3. OmicsBox (v2.0.36) was used for functional annotation of the DEGs with Gene Ontology (GO) and Kyoto Encyclopaedia of Genes and Genomes (KEGG) databases [12]. Pathway enrichment analysis was performed with Fisher's exact test at *p*-value < 0.05.

## 3. Results

### 3.1. Transcriptomics Data

The statistics of *C. rotundicauda* RNA-seq data are shown in Table 1. A total of 232,628,086 (CrNS) and 245,448,176 (CrLPS) raw reads were generated from high-throughput sequencing. After data cleaning, a total of 190,326,253 (81.83%) high-quality reads of CrNS and 201,180,020 (81.93%) of CrLPS were mapped.

**Table 1.** Statistics of the horseshoe crab *C. rotundicauda* RNA-seq data.

| Sample | Number of Raw Reads | Number of Clean Reads | Clean Bases (Gb) | Mapped Reads | Mapped Ratio (%) |
|---|---|---|---|---|---|
| CrNS1 | 73,148,414 | 73,144,174 | 10.92 | 60,165,568 | 82.3 |
| CrNS2 | 76,737,552 | 76,732,692 | 11.46 | 62,257,906 | 81.1 |
| CrNS3 | 82,742,120 | 82,736,822 | 12.36 | 67,902,779 | 82.1 |
| **Total** | **232,628,086** | **232,613,688** | **34.74** | **190,326,253** | **Average: 81.83** |
| CrLPS1 | 82,252,646 | 82,246,900 | 12.27 | 67,152,864 | 81.6 |
| CrLPS2 | 74,186,052 | 74,180,514 | 11.07 | 60,499,561 | 81.6 |
| CrLPS3 | 89,009,478 | 89,003,364 | 13.29 | 73,527,595 | 82.6 |
| **Total** | **245,448,176** | **245,430,778** | **36.63** | **201,180,020** | **Average: 81.93** |

A summary of RNA-seq data of the number of expressed genes in CrNS and CrLPS samples is shown in Figure 1. A total of 10,984 genes were expressed in both CrNS and CrLPS samples and the number of uniquely expressed genes for CrNS and CrLPS was 242 and 149, respectively.

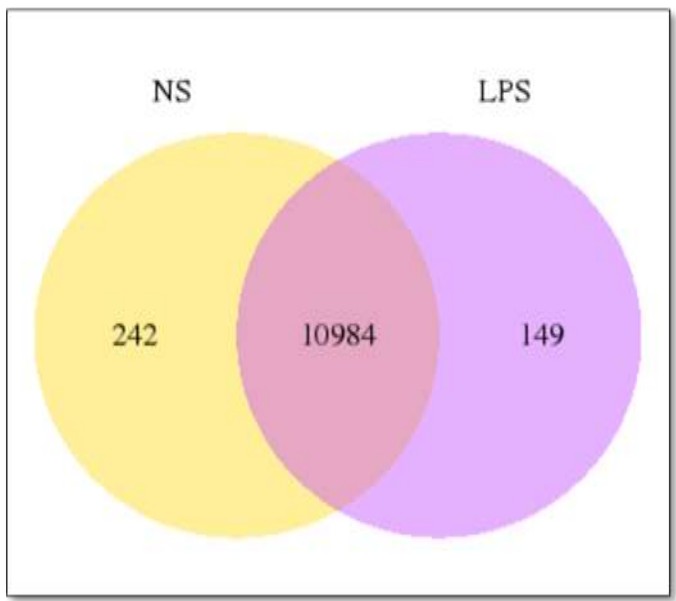

**Figure 1.** Summary of RNA-seq data. Venn diagram showing the number of expressed genes (normalized expression values > 1) in *C. rotundicauda* non-stimulated (NS) and LPS-stimulated (LPS) samples.

The raw RNA-seq reads were deposited in NCBI, Sequence Read Archive (SRA) database under the following accession numbers: CrNS1: SRR14663353, CrNS2: SRR14663352, CrNS3: SRR14663351, CrLPS1: SRR14663350, CrLPS2: SRR14663349, CrLPS3: SRR14663348.

### 3.2. Differentially Expressed Genes (DEGs)

A total of 344 DEGs, consisting of 79 significantly upregulated and 265 downregulated genes, were detected in CrLPS at $p$-value < 0.05 compared with CrNS (Table S1). Out of these, 29 upregulated genes had log-fold change of >0.3 and 154 downregulated genes had log-fold change of <−0.3 (Table S1).

### 3.3. Functional Analysis

GO term analysis showed 155 enriched GO terms (Table S2). Figure 2 shows the top 30 most enriched GO terms of *C. rotundicauda* DEGs. The enriched GO terms DEGs were associated with all GO functional categories [12]: biological process (100), cellular component (21), and molecular function (32) (Table S2, Figure 2). Terms associated with the immune function, such as cytoskeleton and structural constituent of cytoskeleton, were enriched (Table S2).

Figure 3 shows the results of the KEGG enriched pathway analysis of *C. rotundicauda* DEGs. Several pathways related to immune defence, such as Toll and Imd signalling pathway, apoptosis, MAP kinase (MAPK) signalling pathway, inositol phosphate metabolism, phagosome and leucocyte endothelial migration, leucocyte endothelial migration, and infection with gram-negative bacteria, were enriched. Pathways related to infections, such as bacterial invasion of epithelial cells, viral myocarditis, malaria, leishmaniasis, and bacterial infections (*Escherichia coli*, *Salmonella*, *Shigella*, *Yersinia*, and *Pertussis*), were also enriched.

Among the DEGs associated with cytoskeleton function were the following: zyxin, actin-like protein, tubulin alpha-1C chain, CAP-Gly domain-containing linker protein 2, and integrin linked kinase (Table S1). After LPS stimulation, other DEGs related to immune defence were found, such as coagulogen, proclotting enzyme, ribosomal protein S6 kinase alpha-5, autophagy related-1, LK6 kinase, importin subunit alpha-3, tachystatin B1, MyD88, galectin-B, and histone H2A, among others (Table S1).

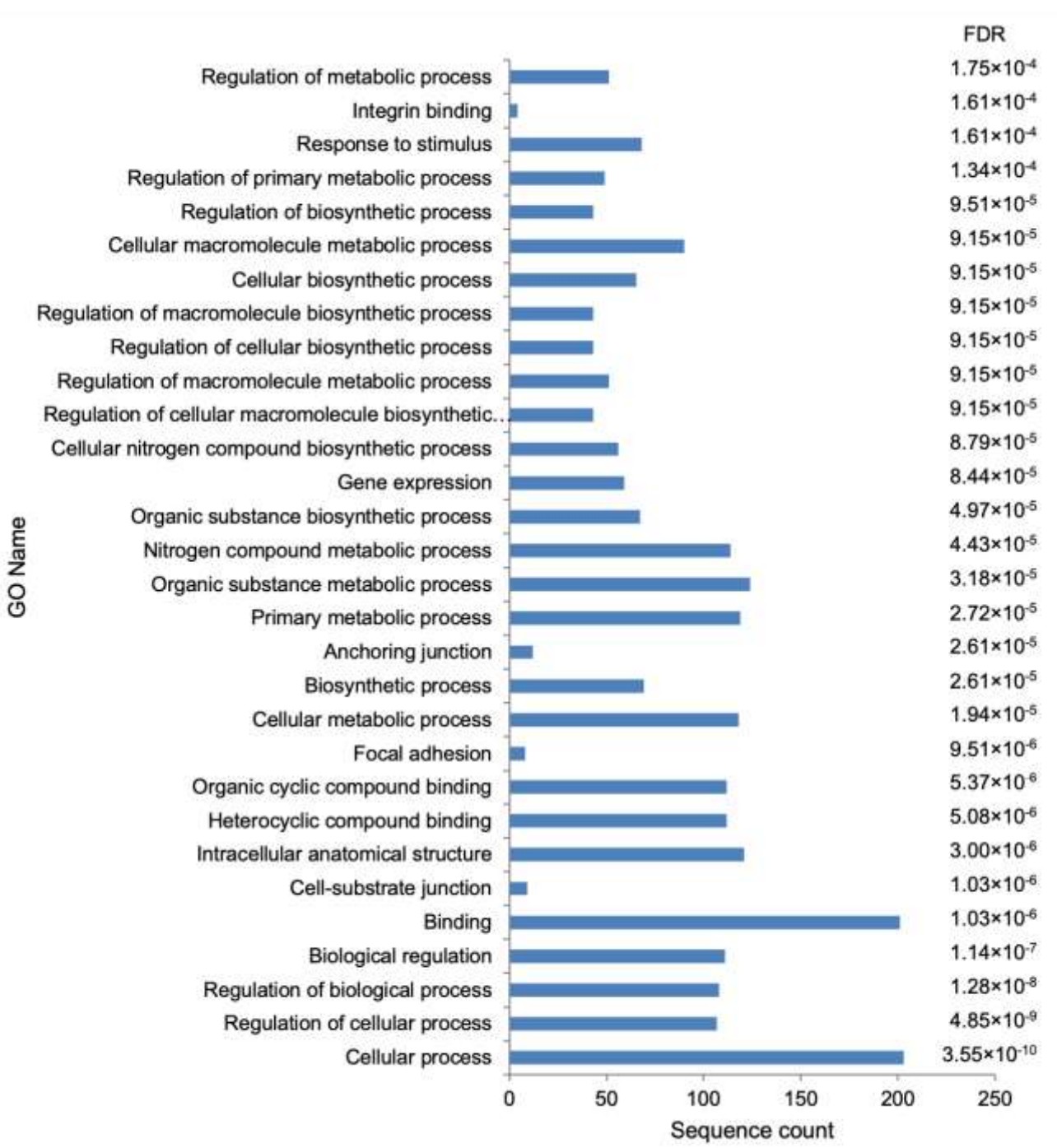

**Figure 2.** The top 30 most enriched gene ontology (GO) terms of *C. rotundicauda* differentially expressed genes (DEGs). Sequence count and false discovery rate (FDR) are shown.

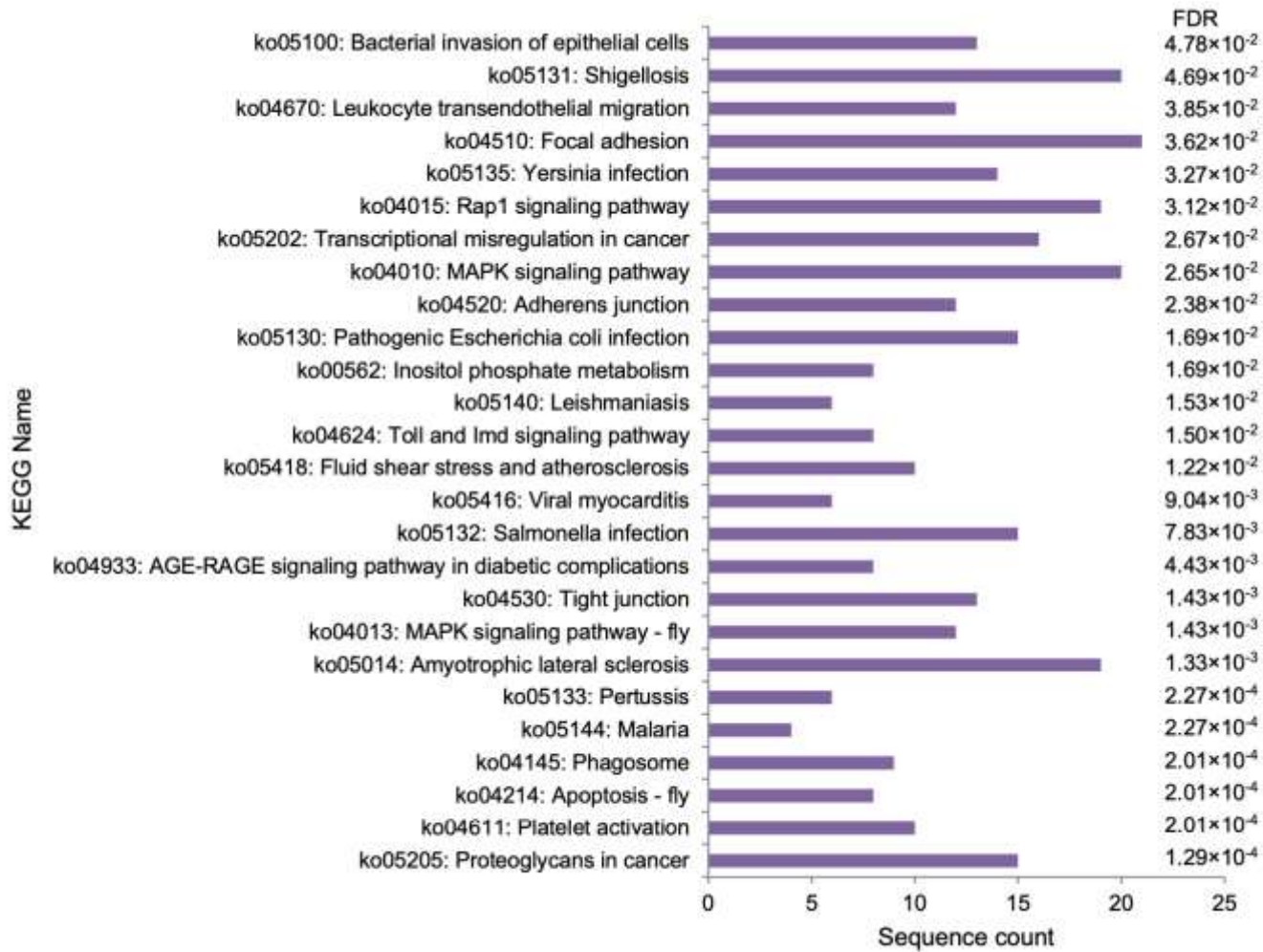

**Figure 3.** Enriched KEGG pathways of *C. rotundicauda* differentially expressed genes (DEGs). Sequence count and false discovery rate (FDR) are shown.

## 4. Discussion

HSCs have survived through time by depending mainly on the innate immune system through different mechanisms, including hemolymph coagulation, encapsulation, melanisation, phenol oxidase activation, cell agglutination, reactive oxygen species, and phagocytosis [13,14]. In addition to these mechanisms, HSCs release defence molecules stored in secretory granules such as antibacterial substances, serine protease zymogens, coagulogen, protease inhibitors, antimicrobial peptides, and lectins, among others [13–15]. In addition to the dominant role of the innate immune defence system rudimentary specific immune responses such as the down syndrome cell adhesion molecule (DSCAM) system are also present, as has previously been described in arthropods and other invertebrates [16–18]. Hemocytes are the pivotal elements in the immune defence of HSCs, activated through the recognition of pathogen-associated molecular patterns (PAMPs) by pattern-recognition receptors (PRRs), which trigger full activation of immune defence mechanisms [13,19,20]. LPS is present in the cell wall of gram-negative bacteria and is the most important stimulus for HSC hemocyte immune activation [13,21–23].

Previous studies have reported on the expression of immune response genes in *T. tridentatus* and *C. rotundicauda* after the infection with gram-negative microorganisms [5,15]. Transcriptomic profiles of *T. gigas* isolated hemocytes after LPS stimulation have also been reported [7]. To help further understand the mechanisms of innate immune defense in HSCs, the gene expression of *C. rotundicauda* hemocytes upon direct stimulation with LPS was explored by implementing an ex vivo high-throughput transcriptomic sequencing.

Multiple GO terms related to all functional categories and KEGG pathways were enriched. Many of them are related to immune activation, showing the high impact of LPS exposure on cellular activation.

In a recent study, we found the enrichment of similar GO terms, such as cellular process, intracellular anatomical structure, cellular metabolic process, and nitrogen compound metabolic process, among others, upon stimulation of *T. gigas* hemocytes with LPS [7] (Table 2). Further, comparison with the DEGs obtained from *T. tridentatus* following infection with a gram-negative microorganism also showed enrichment of similar GO terms (cellular process, regulation of cellular process, regulation of biological process, biological regulation, binding, focal adhesion, and response to stimulus [15]) (Table 2).

**Table 2.** The top 30 most enriched gene ontology (GO) terms of *C. rotundicauda* (current study) differentially expressed genes (DEGs) compared with *T. gigas* [7] and *T. tridentatus* [15].

| No. | Annotation | *C. rotundicauda* (Current Study) | *T. gigas* | *T. tridentatus* |
|---|---|---|---|---|
| 1 | GO:0009987 | cellular process | X | X |
| 2 | GO:0050794 | regulation of cellular process | | X |
| 3 | GO:0050789 | regulation of biological process | | X |
| 4 | GO:0065007 | biological regulation | | X |
| 5 | GO:0005488 | binding | | X |
| 6 | GO:0030055 | cell–substrate junction | | |
| 7 | GO:0005622 | intracellular anatomical structure | X | |
| 8 | GO:1901363 | heterocyclic compound binding | | |
| 9 | GO:0097159 | organic cyclic compound binding | | |
| 10 | GO:0005925 | focal adhesion | | X |
| 11 | GO:0044237 | cellular metabolic process | X | |
| 12 | GO:0009058 | biosynthetic process | | |
| 13 | GO:0070161 | anchoring junction | | |
| 14 | GO:0044238 | primary metabolic process | | |
| 15 | GO:0071704 | organic substance metabolic process | | |
| 16 | GO:0006807 | nitrogen compound metabolic process | X | |
| 17 | GO:1901576 | organic substance biosynthetic process | | |
| 18 | GO:0010467 | gene expression | | |
| 19 | GO:0044271 | cellular nitrogen compound biosynthetic process | | |
| 20 | GO:2000112 | regulation of cellular macromolecule biosynthetic process | | |
| 21 | GO:0060255 | regulation of macromolecule metabolic process | | |
| 22 | GO:0031326 | regulation of cellular biosynthetic process | | |
| 23 | GO:0010556 | regulation of macromolecule biosynthetic process | | |
| 24 | GO:0044249 | cellular biosynthetic process | | |
| 25 | GO:0044260 | cellular macromolecule metabolic process | | |
| 26 | GO:0009889 | regulation of biosynthetic process | | |
| 27 | GO:0080090 | regulation of primary metabolic process | | |
| 28 | GO:0050896 | response to stimulus | | X |
| 29 | GO:0005178 | integrin binding | | |
| 30 | GO:0019222 | regulation of metabolic process | | |

Note: 'x' denotes similar enrichment of the GO terms in previous studies compared with the current findings.

GO terms related to immune function such as cytoskeleton and structural constituent of cytoskeleton enriched in the current study were also similarly enriched in LPS-stimulated *T. gigas* hemocytes [7]. Enrichment of other cytoskeleton-related GO terms (synapse, synapse part and structural molecule activity) was also reported after the infection of *T. tridentatus* with gram-negative bacteria [15]. The cytoskeleton function is implicated in various important processes of HSC protective responses, including phagocytosis and exocytosis of a wide array of defensive molecules upon interaction with LPS [13,24].

KEGG pathway analysis showed a wide spectrum of DEGs associated with immune mechanisms, indicating that multiple immune-related pathways were enriched as a result of LPS stimulation. Toll and Imd signalling pathway was enriched, as has been reported in studies of LPS stimulation of other HSC species [7,15] (Table 3). Toll-like receptors (TLRs) are one of the most important PRRs [25]. The Toll signalling pathway was first described in *Drosophila*, which is of paramount importance in the defence against microbial infection [26]. This pathway has been conserved in evolution and is a key element in the response to LPS [25,26].

**Table 3.** Enriched KEGG pathways of *C. rotundicauda* (current study) differentially expressed genes (DEGs) compared with *T. gigas* [7] and *T. tridentatus* [15].

|  | Annotation | *C. rotundicauda* (Current Study) | *T. gigas* | *T. tridentatus* |
|---|---|---|---|---|
| 1 | ko05100 | Bacterial invasion of epithelial cells |  |  |
| 2 | ko04510 | Focal adhesion |  | X |
| 3 | ko04670 | Leukocyte transendothelial migration |  |  |
| 4 | ko05131 | Shigellosis |  |  |
| 5 | ko04015 | Rap1 signaling pathway |  |  |
| 6 | ko05135 | Yersinia infection |  |  |
| 7 | ko05202 | Transcriptional misregulation in cancer |  | X |
| 8 | ko04520 | Adherens junction |  |  |
| 9 | ko04010 | MAPK signaling pathway |  | X |
| 10 | ko05130 | Pathogenic *Escherichia coli* infection |  |  |
| 11 | ko00562 | Inositol phosphate metabolism |  |  |
| 12 | ko05140 | Leishmaniasis |  | X |
| 13 | ko04624 | Toll and Imd signaling pathway | X | X |
| 14 | ko05418 | Fluid shear stress and atherosclerosis |  | X |
| 15 | ko05416 | Viral myocarditis |  |  |
| 16 | ko04933 | AGE-RAGE signaling pathway in diabetic complications |  |  |
| 17 | ko05132 | Salmonella infection |  |  |
| 18 | ko04530 | Tight junction |  | X |
| 19 | ko04013 | MAPK signaling pathway-fly |  |  |
| 20 | ko05014 | Amyotrophic lateral sclerosis |  |  |
| 21 | ko05133 | Pertussis |  | X |
| 22 | ko05144 | Malaria |  |  |
| 23 | ko04145 | Phagosome |  |  |
| 24 | ko04214 | Apoptosis-fly |  |  |
| 25 | ko04611 | Platelet activation |  | X |
| 26 | ko05205 | Proteoglycans in cancer |  |  |

Note: 'x' denotes similar enrichment of the elements of KEGG observed in previous studies compared with the current findings.

Apoptosis was another enriched pathway in our study, as well as after the infection of *C. rotundicauda* with *Pseudomona aeruginosa* (*P. aeruginosa*) [5] (Table 3). Apoptosis is activated during infection by the invading microorganisms as a virulence mechanism or by the host cell to clear the infection [27,28]. In general, apoptosis is a mechanism that favours the control of infection by the host, and its inhibition is associated with the multiplication and dissemination of microorganisms [28]. In *C. rotundicauda,* during the infection with *P. aeruginosa*, some pro-apoptotic genes (example for COX-1) were down-regulated, probably as a bacterial escape mechanism, whereas other pro-apoptotic genes including one that encodes for amine oxidase were upregulated, possibly as a host response to control the infection [5].

MAPK signalling pathway was also enriched after LPS stimulation. Similar enrichment was also reported after *T. tridentatus* challenge with *Vibrio parahaemolyticus* [15] (Table 3). MAPK signaling pathways are evolutionarily highly conserved and ubiquitously expressed. They are involved in diverse cellular functions including cell proliferation, differentiation, apoptosis and stress responses [29]. The immune response is one of several key functions regulated by MAPKs, with the production of several cytokines, as a consequence of the activation of p38 MAPK, JNK, and ERK pathways [29–31]. LPS is one of the activators of MAPKs after the interaction with TLRs, but this activation is controlled by mechanisms such as the activity of the MAPK phosphatase dual specificity phosphatase 1 (DUSP1), an essential endogenous regulator of the inflammatory response to LPS [32–34].

The inositol phosphate metabolism pathway was found to be enriched upon LPS stimulation (Table 3). Inositol compounds play important roles in the signalling cascades induced by LPS [25,35,36]. The crosstalk between inositol phosphate metabolism and the MAPK pathway has been reported in the immune response to infections, vaccination, and cancer, among other biological processes [37–39].

Phagosome pathway enrichment in our study represents one of the most important components of the phagocytic process, one of the key elements of the immune defence of HSCs [13,40–42] (Table 3). Leucocyte endothelial migration was found to be enriched in LPS-stimulated hemocytes. In this regard, it is important to note that infiltration of hemocytes at injury sites is an important process for the HSCs' homeostasis and defence [23,43]. Hence, considering that LPS is one of the most important elements associated with infections, upregulation of genes associated with cell mobility under the influence of LPS is plausible. In concordance with this result is the enrichment of GO terms associated with the cytoskeleton function previously discussed.

In our study, all enriched pathways belonging to bacterial infections were related to gram-negative microorganisms. It is important to note that gram-negative bacteria have in common the presence of LPS [44,45]. The pathway of pertussis infection was also reported to be enriched on *T. tridentatus* after infection with an LPS-producing bacteria [15].

Many of the DEGs related to immune defence found in the current study were also reported after LPS stimulation of isolated *T. gigas* hemocytes (coagulogen, proclotting enzyme, ribosomal protein S6 kinase alpha-5, autophagy related-1, LK6 kinase, importin subunit alpha-3, zyxin, actin-like protein, tubulin alpha-1C chain, CAP-Gly domain-containing linker protein 2, and integrin linked kinase) [7]. In concordance with our results, coagulogen, proclotting enzyme, MyD88, tachystatin-B1, and galectin-B were found among the DEGs after the infection of *T. tridentatus* with *V. parahaemolyticus* [15], while coagulogen and histone H2A were differentially expressed in *C. rotundicauda* challenged with *P. aeruginosa* [5].

Lipid A, the stimulatory subregion of LPS, represents an important PAMP. LPS is considered a universal prototype of PAMP in mammals. It is not recognized by PPRs of several invertebrate species, including arthropods such as *Drosophila megalogaster*. However, HSCs are extremely sensitive to LPS stimulation [46]. As described in *L. polyphemus*, lipid A is detected by the PPR factor C, which triggers potent activation pathways [46]. Upon interaction with LPS in the hemocyte plasma membrane, factor C is auto processed and activated, cleaving factor B which acts upon the pro-clotting enzyme to promote the

conversion of coagulogen to coagulin with the subsequent clotting process, leading to encapsulation and bacterial elimination. Interacting with a TLR, coagulin also activates the NF-κB signalling [46].

Another important outcome of LPS stimulation is the release of biologically active substances from the HSC hemocyte's intracellular large and small granules, which contain defensive molecules such as protease inhibitors, clotting factors, antimicrobial proteins, and tachyplesin, tachistatin, tachicytins, and big defensins, in addition to a wide array of other antimicrobial peptides [8,13,47–50].

The HSC infection with gram-negative LPS-producing bacteria activates the expression of genes related to multiple signaling pathways such as MAPK, NF-κB, JAK-STAT, C-type lectin receptor, and Toll, among others, as well as coagulation and complement cascades [5,15]. The comparison of our results with other studies showed some differences in the DEGs between the various species of HSCs. In particular, comparison with the study of *T. gigas* using the same methodology [7] revealed some differences in enriched gene ontology (GO) terms, KEGG pathways, and DEGs in response to LPS. This may suggest that different genetic characteristics and habitat could be associated with differences in the immune response. Factors such as species, microbiome composition, diet, ocean acidification, and geographical ubication, among other factors, have been associated with differences in the genome, transcriptome, and immune parameters of HSCs [51–55].

However, these differences should be taken with caution, considering that some of the studies were carried out "in vivo" after stimulation with live bacteria and not directly with LPS, and using different time intervals for evaluation [5,15]. Only one previous transcriptomics study by our group, using another HSC species, was carried out using isolated hemocytes challenged with LPS [7]. Nevertheless, interpretation of biological information solely based on mRNA expression should be done with caution as some studies have shown a poor correlation between mRNA and protein expression levels [56,57], which may be influenced by other transcriptional and regulatory parameters [58].

## 5. Conclusions

An in vitro global gene expression profile of *C. rotundicauda* hemocytes after the LPS stimulation is provided in our study. The differential expression of multiple genes related to the immune defence of HSCs, belonging to multiple functional categories and pathways demonstrates the potent biological response elicited by elements associated with infections. Similar elements of the immune system were shared between different HSC species; however, the presence of many other functions that seem unique to certain species indicates that the structure and function of the HSC innate immune system may not be readily translated across different HSC species.

**Supplementary Materials:** The following supporting information can be downloaded at https://www.mdpi.com/article/10.3390/cimb44120399/s1. Table S1: Full list of differentially expressed genes (DEGs) in LPS-challenge *Carcinoscorpius rotundicauda* hemocytes at an adjusted *p*-value of <0.05; Table S2: Enriched Gene Ontology (GO) terms of *Carcinoscorpius rotundicauda* differentially expressed genes (DEGs). Enrichment analysis was performed in OmicsBox with Fisher's exact test ($p < 0.05$).

**Author Contributions:** M.E.S.: Conceptualization, Methodology, Investigation, Formal analysis, Writing—original draft, Writing—review and editing. K.L.C.: Investigation, Formal analysis, Writing—review and editing. N.-S.L.: Formal analysis, Writing—review and editing. N.I.: Methodology, Resources. M.N.N.: Conceptualization, Supervision, Writing—review and editing. A.A.: Conceptualization, Methodology, Investigation, Formal analysis, Writing—original draft, Writing—review and editing. N.S.Y.: Conceptualization, Resources, Supervision, Writing—review and editing, Project administration, Funding acquisition. All authors have read and agreed to the published version of the manuscript.

**Funding:** This research was funded by the Universiti Sains Malaysia (Grant no. 304/PPSP/602002).

**Institutional Review Board Statement:** Not applicable.

**Data Availability Statement:** The RNA-seq data that support the findings of this study are openly available in NCBI Sequence Read Archive (SRA) database at (https://www.ncbi.nlm.gov, accessed on 17 November 2022) with the accession number (CrNS1: SRR14663353, CrNS2: SRR14663352, CrNS3: SRR14663351, CrLPS1: SRR14663350, CrLPS2: SRR14663349, CrLPS3: SRR14663348).

**Acknowledgments:** Authors thank the financial support by the Universiti Sains Malaysia (Grant no. 304/PPSP/602002).

**Conflicts of Interest:** The authors declare no conflict of interest.

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
