# Peer review of "Transcriptomic Signature of Horseshoe Crab Carcinoscorpius rotundicauda Hemocytes’ Response to Lipopolysaccharides"

_cimb, doi:10.3390/cimb44120399_

Round 1
Reviewer 1 Report
This manuscript used bioinformatics combined with RNA-sequencing (performed in an Illumina HiSEQ platform) to investigate the gene expression changes in isolated C. rotundicauda hemocytes challenged with lipopolysaccharides (LPS) for 1 hour. It is lots of workload due to big data analysis. It appears that this manuscript is suitable for publication in this journal with minor revision. I have some comments that need to be addressed before publication.
1. Please add how much total RNA was needed for constructing the sequencing libraries.
2. It is well established that many genes’ mRNA expression levels may not correlate with their protein expression level (please see the following high cited reference), and a gene generally exert its function via it’s protein expression. Please give a brief discussion in this manuscript.
Thanks for the invitation.
Correlation between Protein and mRNA Abundance in Yeast
https://www.ncbi.nlm.nih.gov › articles › PMC83965
by SP Gygi · 1999 · Cited by 4935 — We have determined the relationship between mRNA and protein expression levels for selected genes expressed in ….
Author Response
Response to Reviewer 1 Comments
Comments: This manuscript used bioinformatics combined with RNA-sequencing (performed in an Illumina HiSEQ platform) to investigate the gene expression changes in isolated C. rotundicauda hemocytes challenged with lipopolysaccharides (LPS) for 1 hour. It is lots of workload due to big data analysis. It appears that this manuscript is suitable for publication in this journal with minor revision. I have some comments that need to be addressed before publication.
Response: Thank you for the time devoted to the review of this manuscript and for your suggestions and questions which will improve the quality of the manuscript.
Point 1: Please add how much total RNA was needed for constructing the sequencing libraries.
Response 1: The amount of total RNA needed for constructing the sequencing libraries was 0.4 microgram (line 90).
Point 2: It is well established that many genes’ mRNA expression levels may not correlate with their protein expression level (please see the following high cited reference), and a gene generally exert its function via its protein expression. Please give a brief discussion in this manuscript.
Correlation between Protein and mRNA Abundance in Yeast. https://www.ncbi.nlm.nih.gov› articles › PMC83965 by SP Gygi 1999 Cited by 4935
Response 2: We have added a brief discussion on the correlation between mRNA and protein expression levels, and cited Gygi et al. 1996 and other relevant papers below (lines 303-306).
References:
- Gygi et al. 1999 https://www.ncbi.nlm.nih.gov/pmc/articles/PMC83965/
- Koussounadis et al. 2015 https://www.nature.com/articles/srep10775
- Maier et al. 2009 https://febs.onlinelibrary.wiley.com/doi/full/10.1016/j.febslet.2009.10.036

Reviewer 2 Report
In this paper, the authors performed transcriptomic analysis on isolated C. rotundicauda hemocytes challenged with LPS. However, this work needs to be further analyzed to illustrate the immune response of this HSC under LPS stimulation. I believe the MS is not suitable for publication in current format. I prefer the author to resubmit it after in-depth analysis, and I have following suggestions.
1. In addition of process analysis, the authors didn’t carry out in-depth analysis on the immune response on the LPS stimulation.
2. The authors should make a clear explanation on how this work was different from their team’s previous work on T. gigas (Comparative transcriptome profiling of horseshoe crab Tachypleus gigas hemocytes in response to lipopolysaccharides), expect for the species?
3. Another concern is the selection of sampling time and concentration of LPS stimulation. The authors should explain the experimental basis. Different sampling time and concentration of LPS will significantly affected the results of RNA-seq analysis.
4. The only further analysis of this work was not sold enough. The references used in the Table 2 and Table 3 analysis of the differences of species on GO and KEGG analysis were ancient (1998 and 2001). Back then the sequencing technique and analysis method were immaturity. The comparison among these three species was not accurate.
5. There was some basic information missing, such as in Line 76 the room temperature was not mentioned, etc.
Author Response
Response to Reviewer 2 Comments
Comments: In this paper, the authors performed transcriptomic analysis on isolated C. rotundicauda hemocytes challenged with LPS. However, this work needs to be further analyzed to illustrate the immune response of this HSC under LPS stimulation. I believe the MS is not suitable for publication in current format. I prefer the author to resubmit it after in-depth analysis, and I have following suggestions.
Response: Thank you for the detailed review of this manuscript. Please find below our response to each suggestion. These responses have been included in the revised manuscript
Point 1: In addition of process analysis, the authors didn’t carry out in-depth analysis on the immune response on the LPS stimulation.
Response 1: Different pathways related to the HSC immune response upon LPS stimulation (cytoskeleton, Toll and Imd signalling pathway, apoptosis, MAPK, inositol phosphate metabolism pathway, phagosome pathway & bacterial infections) were discussed and comparison with reported results in HSCs or other species was also included in the manuscript. Analysis of the immune response to the LPS stimulation can be found in the following lines:
Lines 192-199: GO terms related to immune function such as cytoskeleton and structural constituent of cytoskeleton were enriched in the current study… The cytoskeleton function is implicated in various important processes of HSC protective responses, including phagocytosis and exocytosis of a wide array of defensive molecules upon interaction with LPS…
Lines 200-207: Toll and Imd signalling pathway was enriched … which is of paramount importance in the defence against microbial infection... This pathway has been conserved in evolution and is a key element in the response to LPS…
Lines 215-224: Apoptosis was another enriched pathway in our study… Apoptosis is activated during infection by the invading microorganisms as a virulence mechanism, or by the host cell to clear the infection. In general, apoptosis is a mechanism that favours the control of infection by the host, and its inhibition is associated with the multiplication and dissemination of microorganisms…
Lines 225-235: MAP kinase (MAPK) signalling pathway was also enriched after LPS stimulation... The immune response is one of several key functions regulated by MAPKs, with the production of several cytokines, as a consequence of the activation of p38 MAPK, JNK and ERK pathways… LPS is one of the activators of MAPKs after the interaction with TLRs, but this activation is controlled by mechanisms such as the activity of the MAPK phosphatase dual specificity phosphatase 1 (DUSP1), an essential endogenous regulator of the inflammatory response to LPS…
Lines 236-241: Inositol phosphate metabolism pathway was found enriched upon LPS stimulation… Inositol compounds play important roles in the signalling cascades induced by LPS… The cross-talk between inositol phosphate metabolism and MAPK pathway have been reported in the immune response to infections, vaccination and cancer among other biological processes.
Lines 242-250: Phagosome pathway enrichment in our study represents one of the most important components of the phagocytic process, one of the key elements of the immune defence of HSCs… Leucocyte endothelial migration was found enriched on LPS-stimulated hemocytes. In this regard it is important to note that infiltration of hemocytes at injury sites is an important process for the HSCs homeostasis and defence… Hence, considering that LPS is one of the most important elements associated with infections, upregulation of the expression of genes associated with cell mobility under the influence of LPS is plausible. In concordance with this result is the enrichment of GO terms associated with the cytoskeleton function previously discussed…
Lines 251-254: In our study, all the enriched pathways belonging to bacterial infections were related to gram negative microorganisms. It is important to note that gram negative bacteria have in common the presence of LPS…
Lines 261-269: Many of the DEGs related with immune defence found in the current study were also reported after LPS stimulation of isolated T. gigas hemocytes (coagulogen, proclotting enzyme, ribosomal protein S6 kinase alpha-5, autophagy related-1, LK6 kinase, importin subunit alpha-3, zyxin, actin-like protein, tubulin alpha-1C chain, CAP-Gly domain-containing linker protein 2 and integrin linked kinase)…
We have further added a general discussion about the main processes involved in the immune response to LPS stimulation in HSCs is now included in the current version of the manuscript.
Lines 270-288: Lipid A, the stimulatory subregion of LPS, represents an important PAMP. LPS is considered a universal prototype of PAMP in mammals. It is not recognized by PPRs of several invertebrate’s species, including arthropods such as Drosophila megalogaster, however, HSCs are extremely sensitive to LPS stimulation.
As described in L. polyphemus, Lipid A is detected by the PPR factor C which triggers potent activation pathways. Upon interaction with LPS in the hemocyte plasma membrane, Factor C is auto processed and activated, cleaving Factor B which acting upon the pro-clotting enzyme to promote the conversion of coagulogen to coagulin with the subsequent clotting process leading to encapsulation and bacterial elimination. Interacting with a TLR, coagulin also activate the NF-κB signaling.
Another important outcome of LPS stimulation is the release of biologically active substances from the HSC hemocyte´s intracellular large and small granules, which contains defensive molecules such as protease inhibitors, clotting factors, antimicrobial proteins, and tachyplesin, tachistatin, tachicytins, and big defensins, in addition to a wide array of other antimicrobial peptides. The HSC infection with gram negative LPS producing bacteria activates the expression of genes related to multiple signaling pathways, such as MAPK, NF-κB, JAK-STAT, C-type lectin receptor and Toll, among others, as well as coagulation and complement cascades.
Point 2: The authors should make a clear explanation on how this work was different from their team’s previous work on T. gigas (Comparative transcriptome profiling of horseshoe crab Tachypleus gigas hemocytes in response to lipopolysaccharides), expect for the species?
Response 2: The current study on a different species (C. rotundicauda) was conducted to identify differences in the mechanism of innate immunity in HSCs as the immune parameters, the transcriptome and genomic profiles could be influenced by the species, geographical situation, environmental conditions (ocean acidification), microbiota, diet, and other factors. As such, the same experimental design and conditions were applied in this study as with the previous publication on T. gigas. Despite that, similar results were obtained with some enriched GO terms (lines 185-188 and 192-194), KEGG pathways (lines 200-203) and DEGs (lines 261-265), different sets of differentially expressed genes were observed in both species as presented in Tables 2 and 3. Additional explanation is included in lines 290-293 and 296-298.
Point 3: Another concern is the selection of sampling time and concentration of LPS stimulation. The authors should explain the experimental basis. Different sampling time and concentration of LPS will significantly affected the results of RNA-seq analysis.
Response 3: Evaluation of the whole transcriptome in isolated hemocytes under the LPS influence has not been reported except for our previous work with T. gigas.1 In both studies with T. gigas and C. rotundicauda, the study of Ozaki et al.,2 was used as reference for the detailed methodology and optimal conditions including LPS concentration, cell concentration and time of stimulation. A reference to this article is included in line 73.
References:
- Sarmiento, M.E.; Chin, K.L.; Lau, N.S.; Aziah, I.; Ismail, N.; Norazmi, M.N.; Acosta, A.; Yaacob, N.S. Comparative transcriptome profiling of horseshoe crab Tachypleus gigas hemocytes in response to lipopolysaccharides. Fish & shellfish immunology 2021, 117, 148-156
- Ozaki, S. Ariki, S.I. Kawabata. An antimicrobial peptide tachyplesin acts as a secondary secretagogue and amplifies lipopolysaccharide‐induced hemocyte exocytosis FEBS J., 272 (2005), pp. 3863-3871.
Point 4: The only further analysis of this work was not sold enough. The references used in the Table 2 and Table 3 analysis of the differences of species on GO and KEGG analysis were ancient (1998 and 2001). Back then the sequencing technique and analysis method were immaturity. The comparison among these three species was not accurate.
Response 4: There was an error in the citations for Table 2 and Table 3. The T. gigas and T. tridentatus compared were from references Sarmiento et al. 20211 and Wang et al. 20202, respectively. We apologize for the error and confusion.
- rotundicauda, T. gigas and T. tridentatus compared were all sequenced using Illumina platform and functional annotation with GO, and KEGG was also performed with OmicsBox (formerly known as Blast2GO) for all the three species. The comparison of DEG pathways for the three species was valid as the sequencing and analytical approaches for the three species were similar. In Table 3, we eliminated the C. rotundicauda results (last column)
References:
- Sarmiento, M.E.; Chin, K.L.; Lau, N.S.; Aziah, I.; Ismail, N.; Norazmi, M.N.; Acosta, A.; Yaacob, N.S. Comparative transcriptome profiling of horseshoe crab Tachypleus gigas hemocytes in response to lipopolysaccharides. Fish & shellfish immunology 2021, 117, 148-156.
- Wang, W.-F.; Xie, X.-Y.; Chen, K.; Chen, X.-L.; Zhu, W.-L.; Wang, H.-L. Immune Responses to Gram-Negative Bacteria in Hemolymph of the Chinese Horseshoe Crab, Tachypleus tridentatus. Frontiers in immunology 2021, 11, 584808-584808, doi:10.3389/fimmu.2020.584808.
Point 5: There was some basic information missing, such as in Line 76 the room temperature was not mentioned, etc.
Response 5: Some additional information has been included in section 2.2: LPS challenge (lines 72-74). The room temperature (25oC) is mentioned in lines 77 and 80.

Round 2
Reviewer 2 Report
Accept in present form